# Uncertain Dynamic Characteristic Analysis for Structures with Spatially Dependent Random System Parameters

**DOI:** 10.3390/ma16031188

**Published:** 2023-01-30

**Authors:** Wenyi Du, Juan Ma, Changhu Zhou, Yongchun Yan, Peter Wriggers

**Affiliations:** 1Key Laboratory of Electronic Equipment Structure Design, Ministry of Education, Xidian University, Xi’an 710071, China; 2Institute of Continuum Mechanics, Leibniz Universitaet Hannover, An der Universitaet 1, Gebäude 8142, 30823 Garbsen, Germany

**Keywords:** free vibration, random field, Karhunen–Loeve expansion, multi-dimensional kernel density estimation, random dynamic characteristic

## Abstract

This work presents a robust non-deterministic free vibration analysis for engineering structures with random field parameters in the frame of stochastic finite element method. For this, considering the randomness and spatial correlation of structural physical parameters, a parameter setting model based on random field theory is proposed to represent the random uncertainty of parameters, and the stochastic dynamic characteristics of different structural systems are then analyzed by incorporating the presented parameter setting model with finite element method. First, Gauss random field theory is used to describe the uncertainty of structural material parameters, the random parameters are then characterized as the standard deviation and correlation length of the random field, and the random field parameters are then discretized with the Karhunen–Loeve expansion method. Moreover, based on the discretized random parameters and finite element method, structural dynamic characteristics analysis is addressed, and the probability distribution density function of the random natural frequency is estimated based on multi-dimensional kernel density estimation method. Finally, the random field parameters of the structures are quantified by using the maximum likelihood estimation method to verify the effectiveness of the proposed method and the applicability of the constructed model. The results indicate that (1) for the perspective of maximum likelihood estimation, the parameter setting at the maximum value point is highly similar to the input parameters; (2) the random field considering more parameters reflects a more realistic structure.

## 1. Introduction

Structural dynamic characteristics, including the natural frequency and natural mode, as a crucial indicator for the vibrational properties of engineering structures [1], have been widely realized and studied for many years. With the help of finite element method (FEM), the structural dynamic characteristic can be adequately dealt with by investigating the generalized eigenvalue and eigenvector problems [2,3,4]. For example, Gorman and Yu [4] reviewed the method of superposition in vibration analysis of plates and shells, especially focusing on the Gorman method for accurate establishment of eigenvalues and mode shapes in free vibration analysis of rectangular plates. Although the system variables of the concerned structures are broadly accounted for as deterministic, it has been demonstrated that the fluctuation, i.e., uncertainties, of these parameters inevitably and inherently correlated to the structural modelling and analysis process [5,6,7]. The complexity of the actual structural material properties and various random errors during the manufacturing process will result in uncertainty of the structural parameters, such as vibration of the machine tool, random variation in the temperature during processing, etc., which will cause uncertainty among a group of structural components with the same nominal size that are manufactured with the same material and the same processing method [8] and ultimately lead to random fluctuation of material properties around the mean value and a certain correlation between the fluctuation and machining dimension direction [9,10]. In addition, a group of structural members with the same nominal size, because of the uncertainty of their material parameters, may have similar but different dynamic characteristics [11]. The existence of such uncertainties intrinsically has an influence on the believability of the analyzing results of the structural dynamic behaviors [12,13,14]. Hence, it is urgent to develop an uncertain free vibration analysis framework for more effective and meaningful estimation on the structural dynamic characteristics.

In general, uncertain dynamic characteristic analysis is implemented with probabilistic/stochastic approaches, which are based on the theory of probability or statistics [15,16,17,18,19]. Wan [15] used low-order statistical moments to adopt to characterize the uncertainty of modal frequencies of two bridges with assumed normally and uniformly distributed parameters. Liu et al [16] presented a probabilistic boundary element method for analysis of the statistics of structural eigenvalues and eigenvectors with random shape parameters. Consequently, in most of the literature listed above, the relevant uncertainties of the structural parameters are modelled as random discrete variables with predefined statistical information, such as mean values and variance. Over the past decade, stochastic finite element methods (SFEM) have been paid much attention and have been applied to structural analysis of static responses [20,21] or dynamic responses analysis [22,23,24], in which structural response problems were addressed by SFEM by incorporating probabilistic strategies within the FEM. Numerous computational procedures have been developed for solving the random static problems [25,26], as well as other engineering applications, including reliability problems [27]. Normally, there are two categories to implement SFEM: simulation approaches (e.g., the Monte-Carlo method), which are capable of offering the probabilistic features of the concerned structural responses based on the statistics of samples obtained from the simulation [28,29,30]; and non-simulation methods, which approach the statistical characteristics of the structural outputs by carrying out various numerical methods [31]. Development of SFEM in structural engineering, however, has not yet deeply and adequately extended to eigenvalue problems despite their importance in many applications, including the dynamic response of structures.

Moreover, admitting the universal application of SFEM, the creditability of such stochastic processes is conditional to the availability of the statistical information of the concerned uncertainties in practical engineering applications [32]. Especially, the research on characterization of structural uncertainty and modification of structural parameters mainly focuses on expansion of some structural or material parameters, such as elastic modulus, the moment of inertia, thickness, etc., and inputting the mean values and mean variances of these discrete random variables into the structural system for uncertainty analysis. Such constructed models are not very complex and comparatively easy to resolve in most cases but cannot accurately reflect the real uncertainty existing in input parameters and output outcomes of the actual structural system, although various effective methods can be used for stochastic finite element analysis, as listed above. In fact, there are many factors that affect the uncertainty of structural material parameters, resulting in the random distribution of material parameters in the structural space. For example, manufacturing processes can easily lead to spatial variations in the load and material properties, such as moduli and density. When rolling steel plates, the runout of the rolling head presents a trigonometric function law, which will inevitably cause uneven thickness of the structure and then variation in random physical parameters with structural spatial sizes. With the robust progress of uncertainty analysis, SFEM has been escalated with consideration of the spatial dependency of uncertain system parameters by incorporating the theory of random field with structural static analysis [6,10,33,34] or dynamic analysis of simple one-dimensional random field [35,36]. Moreover, there is still a lack of verification methods for SFEM with random field. In most cases, the Monte-Carlo method is used for such a purpose, but it definitely and inevitably has the disadvantage of absolute dependence on the amount of samples [37,38,39,40]. Hence, it is very crucial and urgent to build an effective analysis model of free vibration for the structure with random field and to develop a new verification method regarding the presented model.

Aiming at the uncertainty of structural material parameters, a method used of structural dynamic characteristics analysis and the corresponding verification are presented in the context of stochastic finite element method by incorporating the theory of random field with the finite element method. First, the uncertainty of structural material parameters is represented by the parameter setting model of random field theory, and the uncertainty of structural parameters is quantified with the random field model based on Gauss kernel function. Then, the simulation and discretization of the parameter setting model of random field are implemented with Karhunen–Loeve expansion method, and the structural dynamic characteristics are analyzed in the frame of finite element method, followed by acquisition of the probability distribution density function of the natural frequency by using the multi-dimensional kernel density estimation method. Afterwards, the input parameters of the model are quantified and verified by the maximum likelihood estimation method after comparing the experimental results with the simulation results. Finally, two examples are, respectively, used as one-dimensional and two-dimensional cases of random fields to validate the applicability and effectiveness of the proposed method.

## 2. Representation of the Uncertainty with Random Field

The errors of machining, heat treatment, and material itself may cause uncertainty of the structural system. These errors are usually small and independent. According to central limit theorem, the distribution of many independent and small random variables follow a Gaussian distribution. Actually, the assumption of Gaussian distribution is easy to calculate and the corresponding problems can be solved. In this work, Gauss random field model is used to describe the uncertainty of material parameters of structural system.

### 2.1. Gauss Random Field Model

Gauss random field has two characteristics: (1) its mathematical expectation μ and variance $\sigma^{2}$ are constants independent of position coordinates; i.e., $m_{\omega }(x_{i})=\mu$, $D_{\omega }(x_{i})=\sigma^{2}$, whereby $\omega (x_{i})$ is a random number and $x_{i}$ represents a point in space; (2) its autocovariance function is uniquely related to the relative position distance of two points in the random field but not to the absolute position coordinate of two points; i.e., autocovariance $C_{\omega }\left(x_{i},x_{i}+\tau \right)=C_{\omega }(\tau )=\sigma^{2}\rho (\tau )$, whereby τ is the relative distance between two points, and ρ(τ) is the autocorrelation function of the random field. In addition, another important parameter of Gaussian random field is the correlation distance L, which indicates that the parameters within the correlation distance have obvious correlation. The key to establish a random field is to construct its covariance matrix.

In the framework of finite element, the continuous Gauss random field needs to be discretized into random variable vector for the following structural dynamic analysis. There are several commonly used discretization methods for Gauss random field, i.e., spectrum representation, Karhunen–Loeve expansion (K-L expansion), and so on.

### 2.2. Karhunen–Loeve Expansion

Karhunen–Loeve expansion has been widely applied to the continuous process [31]. Essentially, a random field is decomposed into a series of uncorrelated random variables and certain coefficients, such as eigenfunctions and eigenvalues, by using the K-L expansion. K-L expansion has the following advantages: it has the characteristics of mean square convergence for any type of random field; compared with other discretization methods, when the finite terms of expansion are the same, K-L expansion has the minimum mean square error [6]. In the K-L expansion, a random field H(x,θ) can be expanded into a group of countable and orthogonal random variables; i.e., H(x,θ) can be expanded into a combination of the random scalars $\xi_{n}(\theta )$. The specific development process is as follows [1]:(1)$$H(x,\theta )=\langle E(x)\rangle +\sum_{n=0}^{m}\xi_{n}(\theta )\sqrt{\lambda_{n}}f_{n}(x)$$
whereby x=(x,y) is the coordinate of a point in space, 〈E(x)〉 is the mathematical expectation, θ is a random event, $\xi_{n}{\left(\theta \right)}_{}(n=0,1,\ldots ,m)$ form a Gaussian random sequence with zero mean and they are uncorrelated to each other, and $\lambda_{n}$ and $f_{n}(x)$ are eigenvalues and eigenfunctions of the autocovariance matrix of random field $C(x_{1},x_{2})$.

The key of K-L expansion for a random field is to obtain the eigenvalue and eigenfunction of autocovariance matrix $C(x_{1},x_{2})$. Because the covariance matrix of a random field is defined in the regular geometric space domain, the eigenvalues and eigenvectors can be easily obtained. The detailed solution process of autocorrelation matrix $C(x_{1},x_{2})$ is as follows [5].
(2)$$\int_{D}C\left(x_{1},x_{2}\right)f_{n}(x_{2})dx_{2}=\lambda_{n}f_{n}\left(x_{1}\right)$$ whereby the autocovariance function $C(x_{1},x_{2})$ is bounded, symmetric, and positive definite, which ensures that the eigenvalues and eigenfunctions have the following properties: (1) the set of eigenfunctions $f_{i}\left(x\right)$ is orthogonal and complete; (2) for each eigenvalue $\lambda_{k}$, there are at most a limited number of linearly independent eigenfunctions; (3) there is at most one countable infinite set of eigenvalues; (4) all of the eigenvalues are positive real numbers; (5) the autocovariance function $C(x_{1},x_{2})$ can be decomposed into the following forms:(3)$$C\left(x_{1},x_{2}\right)=\sum_{k=1}^{\infty }\lambda_{k}f_{k}\left(x_{1}\right)f_{k}\left(x_{2}\right)$$

In the case of one-dimensional (1D) random field, the following autocorrelation kernel function can be used for solving the eigenfunction $f_{n}(x)$ in Equation (2)
(4)$$C\left(x_{1},x_{2}\right)=e^{-\left|x_{1}-x_{2}\right|/L}$$
whereby the correlation length of random field L also reflects the attenuation degree of correlation between two points $x_{1}$ and $x_{2}$. Therefore, $C\left(x_{1},x_{2}\right)$ is a function of variable $\left|x_{1}-x_{2}\right|$ and the parameter L; the integral area D in Equation (2) is a real number interval in the case of one-dimensional random field, and it can be taken as D=[−a,a]. Therefore, Equation (2) can be converted into:(5)$$\int_{-a}^{+a}e^{-\left|x_{1}-x_{2}\right|/L}f_{n}\left(x_{2}\right)dx_{2}=\lambda_{n}f_{n}\left(x_{1}\right)$$

Equation (5) is further expanded to obtain:(6)$$\int_{-a}^{x_{1}}e^{-\left(x_{1}-x_{2}\right)/L}f_{n}\left(x_{2}\right)dx_{2}+\int_{x_{1}}^{+a}e^{\left(x_{1}-x_{2}\right)/L}f_{n}\left(x_{2}\right)dx_{2}=\lambda_{n}f_{n}\left(x_{1}\right)$$

Finding the first derivative of Equation (6) to $x_{1}$ yields
(7)$$\lambda_{n}f_{n}^{\prime }\left(x_{1}\right)=-\frac{1}{L}\int_{-a}^{x_{1}}e^{-\left(x_{1}-x_{2}\right)/L}f_{n}\left(x_{2}\right)dx_{2}+\frac{1}{L}\int_{x_{1}}^{+a}e^{\left(x_{1}-x_{2}\right)/L}f_{n}\left(x_{2}\right)dx_{2}$$

Finding the derivative of Equation (7) to $x_{1}$ once again and substituting $x_{1}$ with x, then the differential equation in general form can be obtained as follows
(8)$$\lambda_{n}f_{n}^{\prime \prime }(x)=\left(-\frac{2}{L}+\frac{\lambda_{n}}{L^{2}}\right)f_{n}(x)$$

Let $\omega^{2}=\left(-\frac{2}{L}+\frac{\lambda_{n}}{L^{2}}\right)/\lambda_{n}$; Equation (8) can be converted into
(9)$$f_{n}^{\prime \prime }(x)+\omega^{2}f_{n}(x)=0(-a\leq x\leq a)$$

To solve Equation (9), its boundary conditions should be found at first. Substituting $x_{1}=-a$ and $x_{1}=a$ into Equations (6) and (7), respectively, and rearranging them, the following boundary conditions can be obtained
(10)$$\frac{1}{L}f_{n}(a)+f_{n}^{\prime }(a)=0$$
(11)$$\frac{1}{L}f_{n}(-a)-f_{n}^{\prime }(-a)=0$$

In summary, Equation (2) has been transformed into a general differential Equation (9) with its boundary conditions Equations (10) and (11), and the general solution of Equation (9) is
(12)$$f_{n}(x)=c_{1}\cos(\omega x)+c_{2}\sin(\omega x)$$

By substituting Equation (12) into boundary conditions and then rearranging the equations, this yields:(13)$$c_{1}\left(\frac{1}{L}-\omega \tan(\omega a)\right)+c_{2}\left(\omega +\frac{1}{L}\tan(\omega a)\right)=0,c_{1}\left(\frac{1}{L}-\omega \tan(\omega a)\right)-c_{2}\left(\omega +\frac{1}{L}\tan(\omega a)\right)=0$$

Based on Equation (13), to obtain the solution of differential Equation (9), namely, to obtain Equation (12), the following conditions must be met:(14)$$\frac{1}{L}-\omega \tan(\omega a)=0,\omega +\frac{1}{L}\tan(\omega a)=0$$

Solving Equation (14) and expressing the solution of the first equation as $\omega_{n}^{\#}$, $\omega_{n}^{\#}$ should be within the interval $\left[\left(\frac{n}{2}-\frac{1}{2}\right)\frac{\pi }{a},\frac{n}{2}\cdot \frac{\pi }{a}\right],(n=1,3,\ldots ,m-1)$, whereby m is an even number and it indicates the truncated number during the K-L expansion of Equation (1). Expressing the solution of the second equation of Equation (14) as $\omega_{n}^{*}$, $\omega_{n}^{*}$ is within the interval $\left[\left(\frac{n}{2}-\frac{1}{2}\right)\frac{\pi }{a},\frac{n}{2}\cdot \frac{\pi }{a}\right],(n=2,4,\ldots ,m)$. The corresponding eigenfunctions are the following Equations (15) and (16)
(15)$$f_{n}(x)=\left[\cos\left(\omega_{n}^{\#}x\right)\right]/\sqrt{a+\frac{\sin\left(2\omega_{n}^{\#}a\right)}{2\omega_{n}^{\#}}}\left(n=1,3,\ldots ,m-1\right)$$
(16)$$f_{n}^{*}(x)=\sin\left(\omega_{n}^{*}x\right)/\sqrt{a-\frac{\sin\left(2\omega_{n}^{*}a\right)}{2\omega_{n}^{*}}}\left(n=2,4,\ldots ,m\right)$$

Furthermore, based on the transformation relation $\omega^{2}=\left(-\frac{2}{L}+\frac{\lambda_{n}}{L^{2}}\right)/\lambda_{n}$, the corresponding eigenvalues to $\omega_{n}^{\#}$ and $\omega_{n}^{*}$ can be derived as
(17)$$\lambda_{n}=2/\left[L\cdot {\left(\omega_{n}^{\#}\right)}^{2}+\frac{1}{L}\right]\left(n=1,3,\ldots ,m-1\right)$$
(18)$$\lambda_{n}^{*}=[2/L\cdot {\left(\omega_{n}^{*}\right)}^{2}+\frac{1}{L}]\left(n=2,4,\ldots ,m\right)$$

Based on Equations (14)–(16), take the Karhunen–Loeve expansion of beams for example. Set the beam length l=1m, a=0.5m and the correlation length L=0.3m. When the number of truncations is taken as n=1,2,3,4,5,6, the eigenfunctions $f_{n}\left(x\right)$ are illustrated in the left part of Figure 1. It can be seen directly that the eigenfunctions are actually a series of trigonometric functions, and their periods and amplitudes are related to the values of the chosen order n; that is, the larger the value of n, the smaller the period of $f_{n}\left(x\right)$. Based on Equations (17) and (18), when l=5m, a=0.5m, and L=0.2m,0.3m,0.5m,1m,2m,5m, the eigenvalues are shown in the right part of Figure 1. It can be seen from Figure 1 that the random field H(x,θ) is composed of eigenvalues $\lambda_{n}$ and eigenfunctions $f_{n}\left(x\right)$ based on Equation (1); the larger the value of L in the right part is, the larger the value of low-order eigenvalues, such as $\lambda_{2}$, and the larger the proportion of low-order eigenfunctions, such as $f_{2}\left(x\right)$ in H(x,θ), as shown in the left part, so the fluctuation of random field is more gentle.

After the eigenfunctions and eigenvalues of 1D random field are obtained, the autocorrelation function can be obtained based on Equation (3); i.e., $C\left(x_{1},x_{2}\right)=\sum_{k=1}^{\infty }\lambda_{k}f_{k}\left(x_{1}\right)f_{k}\left(x_{2}\right)$. When the length of 1D random field is 1m and the correlation length L is 0.3 m, $C\left(x_{1},x_{2}\right)$ is displayed in Figure 2. Figure 2a is the autocorrelation kernel function represented by Equation (4), and Figure 2b–d are the $C\left(x_{1},x_{2}\right)$ obtained with K-L expansion, respectively, when the truncation number m is taken as 4, 8, and 12, whereby $C\left(x_{1},x_{2}\right)=\sum_{k=1}^{m}\lambda_{k}f_{k}\left(x_{1}\right)f_{k}\left(x_{2}\right)$.

It can be seen from Figure 2 that $C\left(x_{1},x_{2}\right)$ is more and more close to its autocorrelation kernel function with the increasing m; that is, the more accurate the autocorrelation function is, the more precise the random field simulation is. Therefore, in the following examples in Section 4, the truncation number m is selected as 12 with a synthetical consideration of simulation accuracy and calculation workload.

For a two-dimensional (2D) Gaussian random field, its eigenvalues and eigenfunctions can be expressed by the product of the eigenvalues and eigenfunctions of two 1D random fields [3] as follows
(19)$$\lambda_{n}=\lambda_{n1}^{1D}\cdot \lambda_{n2}^{1D}$$
(20)$$f\left(x\right)\equiv f\left(x,y\right)=f_{n1}\left(x\right)\cdot f_{n2}\left(y\right)$$

Substituting Equations (19) and (20) into Equation (1), the K-L expansion of 2D random field can be obtained
(21)$$H\left(x,y,\theta \right)=\bar{H}\left(x,y\right)+\sum_{n=0}^{M}\xi_{n}(\theta )\sqrt{\lambda_{n1}\cdot \lambda_{n2}}f_{n1}(x)\cdot f_{n2}(y)$$

## 3. Structural Parameters Uncertainty Characterizing and Quantification

Based on the proposed Gaussian random field model, the uncertainty of structural parameters will be characterized and quantified in this section. In the framework of finite element, the Gauss random field of structural parameters is discretized into every grid element, and the random dynamic characteristics of the structure are then calculated. Non-parametric estimation of the structural dynamic characteristics will then be implemented by using the kernel density estimation method so as to obtain the distribution function characteristics of the structural random dynamic characteristics. Finally, the distribution characteristics of the output responses from the test and simulation will be compared with each other, and the distribution parameters of the model will be quantified and verification carried out with maximum likelihood estimation.

### 3.1. The Analysis of Structural Dynamic Characteristics with Random Field

Considering that the Young’s modulus of the structure is a Gaussian random field, the random field is discretized based on Equation (1) and then substituted into the following element stiffness matrix and element consistent mass matrix
(22)$$\left[k^{e}\right]=\int_{\Omega }{\left[B\right]}^{T}\left[D\right]\left[B\right]d\Omega$$
(23)$$\left[m^{e}\right]=\int_{\Omega }\rho {\left[N\right]}^{T}\left[N\right]d\Omega$$
whereby [B] is the geometric matrix, [N] is the shape function matrix; Ω is the integral domain, Ω is the unit length for the bar element and beam element, and Ω is the unit area for the plate and shell elements; ρ is the material density; [D] is the elastic matrix shown in Equation (24) for the beam element, $I_{z}$ is the inertia moment of the beam; [D] is the bending stiffness matrix in Equation (25) for the thin plate element, μ is the Poisson’s ratio of the material.
(24)$$\left[D\right]=EI_{z}$$
(25)$$\left[D\right]=\left[\begin{array}{ccc} D_{p} & \mu D_{p} & 0\\ \mu D_{p} & D_{p} & 0\\ 0 & 0 & \frac{1-\mu }{2}D_{p} \end{array}\right]{,}_{}^{}D_{p}=\frac{Eh^{3}}{12\left(1-\mu^{2}\right)}$$

The total stiffness matrix [K] and the total mass matrix [M] of the structural system can be obtained by assembling element stiffness matrix and the element mass matrix and then substituting the boundary conditions of nodes into them. Generally, the vibration that causes damage to the structural system is low-frequency vibration, so, in the subsequent analysis, only the low-order natural frequency of the system is considered in this work, and the matrix iteration method is used for solution so as to quickly obtain the low-order natural frequency of the system and its corresponding vibration mode [6].

Suppose that the structural system represented by the stiffness matrix [K] and the mass matrix [M] is a positive system with n degrees of freedom and the free vibration equation of the structural system described with the flexibility matrix is as follows:(26)$$\left\{X\right\}+\left[R\right]\left[M\right]\left\{\ddot{X}\right\}=\left\{0\right\}$$

Let the solution of Equation (26) be {x}={X}sinpt. Substituting the solution {x} into Equation (26), the mode equation of system is
(27)$$\left[R\right]\left[M\right]\left\{X\right\}=\frac{1}{p^{2}}\left\{X\right\}$$

Let $\lambda =\frac{1}{p^{2}},\;\left[S\right]=\left[R\right]\left[M\right]$, Equation (27) can be converted into
(28)[S]{X}=λ{X}

For the first-order eigenvalue $\lambda_{1}$, the relation holds; i.e., $\left[S\right]{\left\{X\right\}}_{1}=\lambda_{1}{\left\{X\right\}}_{1}$. Based on this relationship, the matrix iteration method is used to carry out iteration, and the maximal eigenvalue $\lambda_{1}$ and the corresponding eigenvector ${\left\{X\right\}}_{1}$ can be obtained. The specific iteration steps are as follows(1)Taking any normalized mode shape ${\left\{u\right\}}_{0}$ as the initial solution vector and carrying out the first iteration according to the formula $\left[S\right]{\left\{u\right\}}_{0}=a_{1}{\left\{u\right\}}_{1}$, whereby the first components in both ${\left\{u\right\}}_{0}$ and ${\left\{u\right\}}_{1}$ are normalized to 1.(2)If ${\left\{u\right\}}_{1}\neq {\left\{u\right\}}_{0}$, assigning ${\left\{u\right\}}_{1}$ to ${\left\{u\right\}}_{0}$ as the trial solution vector, and repeating step (1) until the k-th iteration meeting $\left[S\right]{\left\{u\right\}}_{k-1}=a_{k}{\left\{u\right\}}_{k}$, it indicates that the iteration converges if ${\left\{u\right\}}_{k-1}={\left\{u\right\}}_{k}$, then $\lambda_{1}=a_{k}$ and ${\left\{X\right\}}_{1}={\left\{u\right\}}_{k}$. It should be noted that the ideal results ${\left\{u\right\}}_{k-1}={\left\{u\right\}}_{k}$ actually cannot be obtained in the actual iteration process due to the existence of errors, but it also means the convergence of iterations when $\left\|{\left\{u\right\}}_{k-1}-{\left\{u\right\}}_{k}\right\|<err$, whereby *err* is the selected error threshold.


To solve the second-order and higher-order eigenvalues and eigenvectors, the dynamic matrix [S] needs cleaning; that is, [S] needs modifying by using the orthogonality of the main mode and the components of the first r-order main modes in [S] should be cleared so as to obtain the r+1 order iterative dynamic matrix. According to the projection theorem of functional theory, the cleaning matrix can be obtained as:(29)$${\left[Q\right]}_{r}=\left[I\right]-\frac{1}{M_{r}}{\left\{X\right\}}_{r}{\left\{X\right\}}_{r}^{T}\left[M\right]$$
whereby $M_{r}={\left\{X\right\}}_{r}^{T}\left[M\right]{\left\{X\right\}}_{r}$. Let the dynamic matrix before and after cleaning be ${\left[S\right]}_{r}$ and ${\left[S\right]}_{r+1}$; the specific process of cleaning is as follows [10]:(30)$${\left[S\right]}_{r+1}={\left[S\right]}_{r}{\left[Q\right]}_{r}={\left[S\right]}_{r}-\frac{\lambda_{r}{\left\{X\right\}}_{r}{\left\{X\right\}}_{r}^{T}\left[M\right]}{M_{r}}$$

The r+1 order eigenvalue of the system $\lambda_{r+1}$ and its corresponding eigenvector ${\left\{X\right\}}_{r+1}$ can be obtained by the operation of cleaning for ${\left[S\right]}_{r+1}$ mentioned above and iterative calculation, and the r+1 order natural frequency of the system can also be obtained, i.e., $f_{r+1}=\frac{1}{2\pi }\sqrt{\frac{1}{\lambda_{r+1}}}$.

### 3.2. Multidimensional Kernel Density Estimation

Maximum likelihood estimation provides a method to evaluate model parameters with given observation data; i.e., by observing the results of many tests, the parameter can be found that can make the probability of sample occurrence the maximum. Since the distribution characteristics of the output response of the model are unknown, it is necessary to make nonparametric estimation for the probability distribution of the output response. Herein, nonparametric estimation for the probability distribution of the output response is implemented with the multi-dimensional kernel density estimation method.

When the structural parameter is random field, the natural frequency $f_{r}$ (r=1,2,…) of the model is also random variable. Suppose that natural frequencies are independent and identically distributed random variables so that the multidimensional kernel density of its distribution density function is estimated as [8]:(31)$$p_{Q}\left(q\right)=\frac{1}{n}\sum_{j=1}^{n}\prod_{k=1}^{m}\left[\frac{1}{h_{k}}K\left(\frac{Q_{k}\left(s_{j}\right)-q_{k}}{h_{k}}\right)\right]$$
whereby $Q_{k}\left(s_{j}\right)={\left[\Phi \right]}^{T}\left(f_{r}-\bar{f_{r}}\right)=\left\{Q_{1}\left(s_{j}\right),Q_{2}\left(s_{j}\right),Q_{3}\left(s_{j}\right),\ldots ,Q_{m}\left(s_{j}\right)\right\}$, $f_{r}=\{f_{r1}\left(s_{j}\right),f_{r2}\left(s_{j}\right),$ and $f_{r3}\left(s_{j}\right),\ldots ,f_{rm}\left(s_{j}\right)\}$ are the sample data of $f_{r}$, $\bar{f_{r}}$ is the mean value of sample data; [C] is the covariance matrix of samples $f_{r}$, [Φ] is the eigenvector matrix of [C]; K(·) is the multivariate kernel function; $h_{k}$ is the window width matrix, the selecting principle of $h_{k}$ is to minimize the mean square error of calculation results; the kernel density estimation points $q=\left(q_{1},q_{2},\ldots ,q_{k},\ldots ,q_{m}\right)$, q is a vector that is the independent variables of kernel density estimation function; herein, q is the vector $Q_{k}\left(s_{j}\right)=$
$\left\{Q_{1}\left(s_{j}\right),Q_{2}\left(s_{j}\right),Q_{3}\left(s_{j}\right),\ldots ,Q_{m}\left(s_{j}\right)\right\}$, which is obtained from the transformation $Q_{k}\left(s_{j}\right)={\left[\Phi \right]}^{T}$ $\left(f_{r}-\bar{f_{r}}\right)$ based on the structural output responses $f_{r}=\{f_{r1}\left(s_{j}\right),f_{r2}\left(s_{j}\right),f_{r3}\left(s_{j}\right),$
$\ldots ,f_{rm}\left(s_{j}\right)\}$.

### 3.3. Maximum Likelihood Estimation

The maximum likelihood estimation is implemented aiming at the output response of the structural model, and the parameter satisfying the probability distribution density is estimated. The parameter with the greatest possibility $\theta^{*}$ is taken as the estimation value of the real parameter θ. Since the log likelihood function is easier to calculate, the kernel density estimation function in Equation (31) is then expressed as
(32)$$\hat{L}\left(\theta \right)=\sum_{i=1}^{num}\ln\left\{\frac{1}{n}\sum_{j=1}^{n}\prod_{k=1}^{m}\left[\frac{1}{h_{k}}K\left(\frac{Q_{}^{\exp}\left(k,i\right)-Q_{\theta }^{\mathrm{mod}}\left(k,j\right)}{h_{k}}\right)\right]\right\}$$
whereby the parameter setting θ=(σ,L), σ is the mean variance of Gaussian random field, and L is the correlation length of random field; simulation output response $Q_{\theta }^{\mathrm{mod}}={\left[\Phi \right]}_{\theta }^{T}\left(f_{\theta }^{\mathrm{mod}}-\bar{f_{\theta }^{\mathrm{mod}}}\right)$, $f_{\theta }^{\mathrm{mod}}$ is the output response of the model with parameter setting θ; the covariance matrix of samples $f_{\theta }^{\mathrm{mod}}$ is ${\left[C\right]}_{\theta }$, and ${\left[\Phi \right]}_{\theta }$ is the eigenvector matrix of ${\left[C\right]}_{\theta }$; $Q_{}^{\exp}={\left[\Phi \right]}_{\theta }^{T}\left(f^{\exp}-\bar{f_{\theta }^{\mathrm{mod}}}\right)$, $f^{\exp}$ is the output responses of tests, and num means the number of test points. According to the assumption of kernel function of multivariate Gaussian distribution [8], the window width is taken as
(33)$$h_{k}=\sqrt{{\left[\lambda \right]}_{kk}}{\left[\frac{4}{n(2+m)}\right]}^{\frac{1}{4+m}}$$
whereby ${\left[\lambda \right]}_{kk}$ is the eigenvalue matrix of covariance matrix ${\left[C\right]}_{\theta }$.

The Gaussian kernel function is chosen as
(34)$$K\left(v\right)=\frac{1}{\sqrt{2\pi }}\exp\left(-\frac{v^{2}}{2}\right)$$

The maximum likelihood estimation method is used to quantify distribution parameters of test samples; i.e., assuming a group of parameters θ, the output responses of parameter samples are taken as the model samples. The probability distribution density function of the output responses is then obtained by using the kernel density estimation aiming at output response values based on Equation (31), and the maximum likelihood estimation function $\hat{L}\left(\theta \right)$ can be computed by inserting the output responses from both test samples and model samples into Equation (32), and, finally, parameter $\theta^{*}$ can be estimated, which corresponds to the maximal value of $\hat{L}\left(\theta \right)$ so as to verify whether it is the input parameter of test samples θ.

## 4. Case Study

### 4.1. I-Beam with One-Dimensional Random Field

Figure 3 is a simply supported steel beam with a section of I-beam, and the length of the beam is 1 m. The inertia moment $I_{z}=245\;{\mathrm{cm}}^{4}$, sectional area $A=14.345\;{\mathrm{cm}}^{2}$, and mass density ρ=11.261 kg/m. The beam is meshed into 60 beam elements, as shown in the right part of Figure 3.

Suppose that Young’s modulus of material E is random field and its mean value is 〈E(X)〉=210 GPa, and the covariance function is $C\left(x_{1},x_{2}\right)=e^{-\left|x_{1}-x_{2}\right|/L}$. Then, K-L expansion of E in every element from Equation (1) is
(35)$$E(x_{i},\theta )=\langle E(X)\rangle +\sum_{n=0}^{m}\xi_{n}(\theta )\sqrt{\lambda_{n}}f_{n}\left(x_{i}\right)\;\left(i=1,2,3,\ldots ,60\right)$$

From Equation (35), the random field E is represented with its mean variance σ and correlation length L; i.e., the parameter setting $\theta_{E}$ is the output parameter of test samples and model samples, $\theta_{{}_{E}}=\left(\sigma_{E},L_{E}\right)$. In the following, the random field E is quantified with numerical tests. Based on Section 3.1, the first eight-order natural frequencies can be solved for every I-beam, and the natural frequency matrix can then be formed as $\left[f\right]=\left\{f_{1},f_{2},f_{3},f_{4},f_{5},f_{6},f_{7},f_{8}\right\}$.

Given an input parameter $\theta_{0}=(\sigma_{0},L_{0})$, and natural frequency matrix of 500 beams ${\left[f\right]}_{8\times 500}^{\exp}$ are considered as the test data samples. Moreover, the other parameter settings $\theta_{ij}=\left(\sigma_{i},L_{j}\right)$
(i=1,2,3,…,10)
j=1,2,3,…,10) are chosen, and the detailed values of parameters are $\left\{\sigma_{1},\sigma_{2},\sigma_{3},\ldots ,\sigma_{10}\right\}=$
{1GPa,2GPa,…,10GPa} and $\left\{L_{1},L_{2},L_{3},\ldots ,L_{10}\right\}=$
{100mm,200mm,300mm,…,
1000mm}. The natural frequencies of 100 beams for each parameter setting $\theta_{ij}$ form a model matrix sample ${\left[f\right]}_{8\times 100}^{\mathrm{mod}}$, and there are $10^{4}$ beams and 100 model matrix samples ${\left[f\right]}_{8\times 100}^{\mathrm{mod}}$ in all. In the following, the kernel density estimation and maximum likelihood estimation will be implemented aiming at natural frequencies, and the identifiability of parameter setting θ=(σ,L) and the effectiveness of random field model will be verified.

#### 4.1.1. Test Data Analysis

For four parameter settings of Young’s modulus $\theta_{E1}=\left(1\mathrm{GPa},100\mathrm{mm}\right)$, $\theta_{E2}=\left(5\mathrm{GPa},100\mathrm{mm}\right)$, $\theta_{E3}=\left(1\mathrm{GPa},1000\mathrm{mm}\right)$, and $\theta_{E4}=\left(5\mathrm{GPa},1000\mathrm{mm}\right)$, the random process distributions of E on 10 beams are demonstrated in Figure 4, in which every curve denotes the value of E in the beam and reflects the randomness of values of E on every point of the beam. From Figure 4, it can be seen that values of the random field E fluctuate around its mean value 210 GPa; the fluctuation along the longitudinal axis reflects the magnitude of variance and the variation of variance with the axial dimension of the beam; just as the meaning of parameter setting $\sigma_{E}$ and $L_{E}^{}$, the influence of $\sigma_{E}$ on the random distribution of E is much greater than that of $L_{E}^{}$. In the case of the same parameter setting used, 10 curves in each figure, i.e., random distribution of E, are very different, but the general distribution law is similar in each figure. Moreover, it can be observed from Figure 4 that $L_{E}^{}$ directly affects the randomness distribution of E on every element in the beam; that is, the larger the correlation length is, the more acute the value fluctuation of E along the direction of beam length.

Based on the conclusions of Figure 4, the parameter setting $\theta_{E}=\left(5\mathrm{GPa},500\mathrm{mm}\right)$ will be used in the subsequent analysis.

#### 4.1.2. One-Dimensional Kernel Density Estimation

In order to verify the applicability of the parameter setting $\theta_{E}^{}=\left(5\mathrm{GPa},500\mathrm{mm}\right)$, the test samples $f_{2}^{\exp}$ and model samples $f_{2}^{\mathrm{mod}}$ of the second-order natural frequency $f_{2}^{}$ of the beam are chosen and used for the one-dimensional kernel density estimation of the single parameter of $f_{2}^{}$.

First, take $L_{E0}=500\mathrm{mm}$ and $\sigma_{E}\in \left\{\sigma_{1},\sigma_{2},\sigma_{3},\ldots ,\sigma_{10}\right\}=\left\{1,2,3,\ldots ,10\right\}\mathrm{GPa}$, which form 10 parameter settings $\theta_{E}^{}$. For 100 beams corresponding to each $\theta_{E}^{}$, $f_{2}^{}$ is solved and the model sample ${\left\{f_{2}\right\}}_{1\times 100}^{\mathrm{mod}}$ is formed. Then, take 100 beams with parameter settings $\sigma_{E0}=5\mathrm{GPa}$ and $L_{E0}=500\mathrm{mm}$ and $f_{2}^{}$ of 100 beams construct test samples ${\left\{f_{2}\right\}}_{1\times 100}^{\exp}$. Figure 5a displays the one-dimensional kernel density estimation of $f_{2}^{}$ calculated based on ${\left\{f_{2}\right\}}_{1\times 100}^{\mathrm{mod}}$ and ${\left\{f_{2}\right\}}_{1\times 100}^{\exp}$. From Figure 5a, it can be seen that distributions of natural frequencies of model samples become more and more concentrated with the decreasing $\sigma_{E}$, and the distributions are very close to each other when $\sigma_{E}$ of both test samples and model samples is 5 GPa.

Moreover, when $\sigma_{E0}=5\mathrm{GPa}$ and $L_{E}\in \left\{L_{1},L_{2},L_{3},\ldots ,L_{10}\right\}=\left\{100,200,300,\ldots ,1000\right\}\mathrm{mm}$, $f_{2}^{}$ of 100 beams form the model samples data ${\left\{f_{2}\right\}}_{1\times 100}^{\mathrm{mod}}$, then taking the test sample data ${\left\{f_{2}\right\}}_{1\times 100}^{\exp}$ when $\sigma_{E0}=5\mathrm{GPa}$ and $L_{E0}=500\mathrm{mm}$, one-dimensional kernel density of $f_{2}$ is estimated based on ${\left\{f_{2}\right\}}_{1\times 100}^{\mathrm{mod}}$ and ${\left\{f_{2}\right\}}_{1\times 100}^{\exp}$, as shown in Figure 5b. From Figure 5b, the distribution of $f_{2}$ computed from model samples becomes more and more concentrated with the decreasing $L_{E0}^{}$; the distributions of $f_{2}$ obtained from model samples and test samples are closest when $L_{E}^{}$ of two types of samples are equal. Once again, Figure 5 shows that the influence of $\sigma_{E}$ on the fluctuation of kernel density is greater than that of $L_{E}^{}$.

In the same way, considering that the Young’s modulus E and mass density ρ are random fields varying with the spatial size of beam, the parameter settings are taken as θ=(5, 500) and 100 beams are used. The 1D kernel density of $f_{2}$ is estimated again, where $\sigma_{E}=5\mathrm{GPa}$ and $\sigma_{\rho }=5\mathrm{kg}/m^{3}$. The left part is the 1D kernel density distribution function estimated of $f_{2}$ when $L_{\rho }=500\mathrm{mm}$ but $L_{E}^{}$ changes, and the right part is the 1D kernel density estimated of $f_{2}$ when $L_{E}=500\mathrm{mm}$ but $L_{\rho }$ varies. When $L_{\rho }=L_{E}=500\mathrm{mm}$, in Figure 5a, $\sigma_{E}$ changes but $\sigma_{\rho }$ = $5\mathrm{kg}/m^{3}$, and Figure 5b is the result estimated when $\sigma_{E}=5\mathrm{GPa}$ but $\sigma_{\rho }$ changes.

It can be seen from Figure 6 that the influence of $\sigma_{E}$ and $\sigma_{\rho }$ on the kernel density distribution function is significantly greater than that of $L_{\rho }$ and $L_{E}^{}$; when other parameters are fixed but $\sigma_{E}$ and $\sigma_{\rho }$ change, respectively, the influence of $\sigma_{E}$ on the kernel density distribution function is greater than that of $\sigma_{\rho }$; when other parameters are fixed but $L_{\rho }$ and $L_{E}^{}$ change, respectively, the influence of $L_{E}^{}$ on the kernel density distribution function of $f_{2}$ is greater. In general, the random field E has a greater influence on the kernel density distribution function of $f_{2}$ than the random field ρ.

#### 4.1.3. Multidimensional Kernel Density Estimation and Maximum Likelihood Estimation

In order to accurately verify the validity of the random field model and the identifiability of the parameter settings, it is necessary to further estimate the multi-dimensional kernel density probability distribution function of the structure output responses, and then to carry out the maximum likelihood estimation to obtain the parameter setting corresponding to the maximal value of likelihood function.

When considering random field E, based on 100 model matrix samples ${\left[f\right]}_{8\times 100}^{\mathrm{mod}}$ from $1\times 10^{4}$ beams and test samples ${\left[f\right]}_{8\times 500}^{\exp}$ from 500 beams, the 8-dimensional kernel density of the first 8-order natural frequencies of the beam is estimated, and then the parameter $\theta_{E}=\left(\sigma_{E},L_{E}\right)$ is estimated by the maximum likelihood based on Equation (32), in which ${\left[f\right]}^{\exp}$ and ${\left[f\right]}_{\theta_{ij}}^{\mathrm{mod}}$ are composed of the first 8-order natural frequencies of the structure. Because the values of probability density function of some points are small, in order to avoid the calculation value of likelihood function being 0, it is necessary to take the logarithm of the value of probability density function first and then implement summation. The obtained log likelihood function is illustrated in Figure 7.

Figure 7a is the log likelihood function $\hat{L}\left(\theta \right)$ of the first 8 natural frequencies when the parameter settings of the test data samples are taken as $\sigma_{E}=5\;\mathrm{GPa},L_{E}=500\;\mathrm{mm}$. It can be seen that, when $\hat{L}\left(\theta \right)$ attains its maximum value, the parameter $\theta^{*}$ of the corresponding point is also (5 GPa,500 mm), which shows that the random field model introduced can identify the model parameter very well, and the random field model is reliable.

In Figure 7b, based on Equation (32), a group of parameter setting values of test samples are randomly taken as $\sigma_{E}=6.8\;\mathrm{GPa},L_{E}=210\;\mathrm{mm}$, and the multi-dimensional kernel density probability distribution function is estimated for the first 8 natural frequencies, and then the maximum likelihood estimation is carried out. The estimated parameter results, that is $\theta^{*}=(7\;\mathrm{GPa},200\;\mathrm{mm})$, are slightly different from the original parameter setting (6.8 GPa,210 mm). This is because the amount of the model samples may not be infinite and the estimation accuracy of the input parameter of test samples is obviously limited by the amount of the model samples, but the peak value of likelihood function can still be obtained around the input parameter θ=(6.8,210).

When considering random fields E and ρ simultaneously, the input parameters of the test samples are taken as two groups, respectively: $\theta_{E}=(\sigma_{E},L_{E})=(5\;\mathrm{GPa},500\;\mathrm{mm})$, $\theta_{\rho }=(\sigma_{\rho },L_{\rho })=$ $\left(250\;\mathrm{kg}/m^{3},500\;\mathrm{mm}\right)$, as well as $\theta_{E}=(\sigma_{E},L_{E})=$
(4 GPa,500 mm), $\theta_{\rho }=(\sigma_{\rho },L_{\rho })=(450\;\mathrm{kg}/m^{3},500\;\mathrm{mm})$. The correlation length of two random fields is fixed; that is, $L_{E}=$
$L_{\rho }=500\;\mathrm{mm}$; the mean variances $\sigma_{E}$ and $\sigma_{\rho }$ are then estimated when the likelihood function $\hat{L}\left(\theta \right)$ is taken as its maximal value. It can be seen from Figure 8 that the parameter setting at the maximum value point of $\hat{L}\left(\theta \right)$ is the same as the input parameters by using the maximum likelihood estimation for the first 8-order natural frequency.

### 4.2. Example 2: Plate with Two-Dimensional Random Field

Figure 9 shows a square steel plate fixed at one end, with a thickness of 0.01 m; the mean values of mass density ρ and Young’s modulus E are $\langle \rho (x,y)\rangle =7850.1\;{\mathrm{kg}/m}^{3}$ and 〈E(x,y)〉=210 GPa. The plate is meshed into 400 rectangular plate elements.

#### 4.2.1. The Expansion and Distribution of Two-Dimensional Random Field

First, considering E is a two-dimensional random field, E is then expanded with K-L expansion as
(36)$$E\left(x_{i},y_{j},\theta \right)=\langle E\left(x,y\right)\rangle +\sum_{n=0}^{M}\xi_{n}(\theta )\sqrt{\lambda_{n1}\cdot \lambda_{n2}}f_{n1}(x_{i})\cdot f_{n2}(y_{j})\left(i=j=1,2,\ldots ,20\right)$$

Taking the parameter setting of random field E, $\theta_{E}=\left(\sigma_{E},L_{Ex},L_{Ey}\right)=\left(5\mathrm{GPa},500\mathrm{mm},500\mathrm{mm}\right)$, as the input parameter of test samples and model samples. When the plate is meshed into different amounts of elements, the distribution of random field E in the plate is displayed in Figure 10.

Moreover, only considering mass density ρ as random field, ρ is expressed with the two-dimensional (2D) K-L expansion as Equation (37), whereby its mean value is $\langle \rho (x,y)\rangle =7800\mathrm{kg}/m^{3}$. Taking its parameter setting as $\theta_{\rho }=\left(\sigma_{\rho },L_{\rho x},L_{\rho y}\right)=\left(7800\mathrm{kg}/m^{3},300\mathrm{mm},300\mathrm{mm}\right)$ and meshing the plate into different number of elements, the distribution of ρ in the steel plate is displayed in Figure 11.
(37)$$\rho \left(x_{i},y_{j},\theta \right)=\langle \rho \left(x,y\right)\rangle +\sum_{n=0}^{M}\xi_{n}(\theta )\sqrt{\lambda_{n1}\cdot \lambda_{n2}}f_{n1}(x_{i})\cdot f_{n2}(y_{j})\;\left(i=j=1,2,\ldots ,20\right)$$

It can be seen from Figure 10 and Figure 11 that the values of E and ρ fluctuate and vary around the mean values 〈E(x,y)〉 and 〈ρ(x,y)〉, and the values of E and ρ randomly distributed along *x* and *y* directions. Moreover, compared with the random distributions of E and ρ plated with 225 meshing elements, the random distributions with 400 elements obviously reflect the real cases better and more accurately.

#### 4.2.2. Kernel Density Estimation and Maximum Likelihood Estimation

When only considering random field E, a group of parameter settings are taken as $\theta_{Eijk}=\left(\sigma_{Ei},L_{Exj},L_{Eyk}\right),(i=1,2,\ldots ,10;\;j=1,2,\ldots ,10;k=1,2,\ldots ,10)$, and the specific parameter values are $\left\{\sigma_{E1},\sigma_{E2},\ldots ,\sigma_{E10}\right\}=\left\{1,2,\ldots ,10\right\}$, $\left\{L_{Ex1},L_{Ex2},\ldots ,L_{Ex10}\right\}=\left\{100,200,\ldots ,1000\right\}$, and $\left\{L_{Ey1},L_{Ey2},\ldots ,L_{Ey10}\right\}=$
{100,200,…,1000}. Taking 100 plates for each parameter $\theta_{Eijk}=\left(\sigma_{Ei},L_{Exj},L_{Eyk}\right)$, the first six natural frequencies of 100 plates are chosen to form the model matrix samples ${\left[f\right]}_{6\times 100}^{\mathrm{mod}}$
$=\left\{f_{1},f_{2},\ldots ,f_{6}\right\}$. A total of 1000 model matrix samples ${\left[f\right]}_{6\times 100}^{\mathrm{mod}}$ and $1\times 10_{}^{5}$ plates are used. Next, kernel density estimation and maximum likelihood estimation will be performed on the test data samples based on the model data samples.

In order to verify the random field model of the plate, parameter settings $\theta_{E1}=\left(5,500,500\right)$, $\theta_{E2}=\left(6,700,300\right)$, $\theta_{E3}=\left(2,700,100\right)$, and $\theta_{E4}=\left(9,650,430\right)$ are, respectively, taken for numerical simulation, and 500 plates are used for computation of each parameter setting. The first six natural frequencies of 500 plates are taken for each parameter $\theta_{E1}=\left(5,500,500\right)$ to form test matrix samples ${\left({\left[f\right]}_{6\times 500}^{\exp}\right)}_{1}$, ${\left({\left[f\right]}_{6\times 500}^{\exp}\right)}_{2}$, ${\left({\left[f\right]}_{6\times 500}^{\exp}\right)}_{3}$, and ${\left({\left[f\right]}_{6\times 500}^{\exp}\right)}_{4}$, respectively, and then the logarithmic likelihood functions $\hat{L}\left(\theta_{E}\right)$ are calculated by Equation (32), and the computational results are listed in Table 1.

From Table 1, according to the point at which $\hat{L}\left(\theta \right)$ attains its maximal value, the parameter setting of test samples, i.e., $\theta_{E}^{*}=\left(\sigma_{E}^{*},L_{Ex}^{*},L_{Ey}^{*}\right)$, can be well estimated by the presented 2D random field model of the plate, which is basically consistent with the input parameters $\theta_{E1}$, $\theta_{E2}$, $\theta_{E3}$, and $\theta_{E4}$ of the test samples. Hence, the constructed 2D random field model is reliable. Similar to the 1D random field model of I-beam, the amount of model sample groups may not be infinite, and the estimation accuracy of test sample parameter is limited by the amount of the model samples. When the maximum likelihood estimation method is used to estimate the test samples with input parameter $\theta_{E4}=\left(9\mathrm{GPa},430\mathrm{mm},650\mathrm{mm}\right)$, the input parameter of test samples can still be estimated comparatively accurately; i.e., $\theta_{E4}^{*}=\left(9\mathrm{GPa},400\mathrm{mm},700\mathrm{mm}\right)$.

Furthermore, when considering random fields E and ρ simultaneously, their parameter settings are, respectively, taken as $\theta_{E}=\left(\sigma_{E},L_{Ex},L_{Ey}\right)=$
$\left(\sigma_{E},500\mathrm{mm},500\mathrm{mm}\right)$ and $\theta_{\rho }=\left(\sigma_{\rho },L_{\rho x},L_{\rho y}\right)=(250\mathrm{kg}/m^{3},$
500mm, 500mm). When $\sigma_{E}$ is taken as 1GPa, 2GPa, …, 10GPa, respectively, parameter settings of the test samples are taken as $\theta_{{}_{E0}}^{}=\left(5\mathrm{GPa},500\mathrm{mm},500\mathrm{mm}\right)$ and $\theta_{{}_{\rho 0}}^{}=\left(250\mathrm{kg}/m^{3},500\mathrm{mm},500\mathrm{mm}\right)$; the 2D kernel density distribution function of the second-order natural frequency $f_{2}$ of test samples is estimated as shown in the left part of Figure 12.

Similarly, parameter settings of E and ρ are $\theta_{E}=\left(\sigma_{E},L_{Ex},L_{Ey}\right)=\left(5\;\mathrm{GPa},500\;\mathrm{mm},\;500\mathrm{mm}\right)$ and $\theta_{\rho }=\left(\sigma_{\rho },L_{\rho x},L_{\rho y}\right)=\left(\sigma_{\rho },500\;\mathrm{mm},500\;\mathrm{mm}\right)$, and, respectively, taking $\sigma_{\rho }$ as $50\;\mathrm{kg}/m^{3}$, $100\;\mathrm{kg}/m^{3}$, …, $500\;\mathrm{kg}/m^{3}$ for computing, and then taking parameter settings of test samples $\theta_{{}_{E0}}^{}=(5\;\mathrm{GPa},500\;\mathrm{mm},500\;\mathrm{mm})$ and $\theta_{{}_{\rho 0}}^{}=\left(250\;\mathrm{kg}/m^{3},500\;\mathrm{mm},500\;\mathrm{mm}\right)$, the 2D kernel density distribution function of $f_{2}$ is estimated as shown in the right part of Figure 12.

When taking input parameter settings of test samples $\theta_{{}_{E0}}^{}=(5\;\mathrm{GPa},\;500\;\mathrm{mm},$
500 mm) and $\theta_{{}_{\rho 0}}^{}=$
$\left(250\;\mathrm{kg}/m^{3},500\;\mathrm{mm},500\;\mathrm{mm}\right)$, the curve of kernel density estimation of $f_{2}$ and the histograms of probability density of $f_{2}$ are illustrated in Figure 13.

Figure 12 shows again that random field E has a greater influence on the kernel density distribution function of structural natural frequency than ρ does. In Figure 13, the curve of kernel density estimations and the histograms of $f_{2}$ agree well, and it is obvious that the curve and histogram in the right part are more reasonable than in the left part with the increasing amount of meshing elements.

In addition, fixing mean variance of parameter $\theta_{E}$, $\sigma_{E}=5\mathrm{GPa}$, and, taking $L_{Ex}$ and $L_{Ey}$ as variables, the variation in log likelihood function $\hat{L}\left(\theta \right)$ of natural frequency with $L_{Ex}$ and $L_{Ey}$ is shown in Figure 14a. Similarly, $\hat{L}\left(\theta \right)$ is obtained in Figure 14b when parameter settings of test samples $\sigma_{E}=2.1\mathrm{GPa},L_{E}=L_{Ex}=L_{Ey}=300\mathrm{mm}$. In Figure 14, the plate is meshed into 400 elements.

It can be seen from Figure 14 that the parameter settings $\theta_{E}^{*}$ obtained corresponding to the maximal value $\hat{L}\left(\theta \right)$ are, respectively, $\theta_{E}^{*}=(\sigma_{e}^{*},L_{Ex}^{*},L_{Ey}^{*})=(5\mathrm{GPa},500\mathrm{mm},500\mathrm{mm})$ and (2GPa,300mm,300mm); based on the multi-dimensional kernel density estimation, parameter settings of test samples can be accurately estimated when the log likelihood function attains its maximal value and the estimated parameter settings are very close to the input parameters of the test samples, which verifies the validity of the constructed model.

In the same way, when only ρ is random field and the input parameter settings $\theta_{\rho }$ of the test samples are $\sigma_{\rho }=500\;\mathrm{kg}/m^{3},L_{\rho x}=400\;\mathrm{mm},L_{\rho y}=700\;\mathrm{mm}$ and $\sigma_{\rho }=150\;\mathrm{kg}/m^{3},L=L_{\rho x}=L_{\rho y}=600\;\mathrm{mm}$, respectively, $\hat{L}\left(\theta \right)$ obtained based on the first six natural frequencies of the plate is displayed in Figure 15.

#### 4.2.3. Investigation of the Random Characteristics

Figure 16 shows the distributions of the lower bound, mean value, and upper bound of the first two-order random natural modes in the steel plate when only considering random field E, and the parameter setting after K-L expansion is taken as $\theta_{E}=\left(\sigma_{E},L_{Ex},L_{Ey}\right)=(5\;\mathrm{GPa},500\;\mathrm{mm},$
500 mm).

When considering random fields E and ρ simultaneously and taking parameter settings $\theta_{E}=(\sigma_{E},$
$L_{Ex},L_{Ey})=\left(5\mathrm{GPa},500\mathrm{mm},500\mathrm{mm}\right)$ and $\theta_{\sigma }=\left(\sigma_{\sigma },L_{\rho x},L_{\rho y}\right)=(250\mathrm{kg}/m^{3},500\mathrm{mm},500\mathrm{mm})$, the distributions of the lower bound, mean value, and upper bound of the first two random natural modes are illustrated in Figure 17.

The mean values $\langle \left\|\Phi_{i}\right\|\rangle$ and mean variances $\sigma_{\left\|\Phi_{i}\right\|}$ of norms of the first 4-order natural modes are computed and listed in Table 2 for different random models so as to compare the influences of different random cases on the structural random natural modes.

From Figure 16 and Figure 17 and the results in Table 2, it can be seen that mean values of random natural modes only considering random field E are very close to those simultaneously considering random fields E and ρ, but mean variances for these two random models are very different and mean variances only considering the randomness of E are obviously smaller than those considering the randomness of E and ρ simultaneously.

The mean values $\langle f_{i}\rangle$ and mean variances $\sigma_{f_{i}}$ of the first four natural frequencies are computed and listed in Table 3 for different random models so as to compare the influences of different random cases on the structural random natural frequencies.

Similarly, investigation of the influences of different random models on random natural frequencies is implemented and the corresponding results are illustrated in Figure 18. From Figure 18 and Table 3, once again, the mean variances and value ranges of random natural frequencies simultaneously considering the randomness of E and ρ are obviously greater than those only considering the randomness of E, and the former reflects the more realistic case in structural engineering than the latter does.

## 5. Conclusions

In this work, an investigation on the stochastic free vibration problem of engineering structures considering material uncertainties is presented. As a novel extension of the conventional uncertain eigenvalue problem, spatially dependent stochastic parameters and random field theory are combined into a numerical analysis framework of stochastic finite element method, and the verification method to validate the proposed parameter setting model and stochastic free vibration model is presented and updated by using the maximum likelihood method:(1)The parameter setting model based on random field theory can represent the spatially dependent uncertainty of structural parameters well, and the parameter setting model presented can describe the randomly varying characteristics of actual structural parameters.(2)The example shows that the parameter settings of the model can be quantified by the output response of the structural system; i.e., structural dynamic characteristics, such as the structural natural frequency, and the mean variance and autocorrelation distance of the parameter of the structure can also be obtained, which is very important to application of random field in engineering.(3)The proposed method can be extended to apply to other structural parameters and can also be used to establish and quantify the parameter setting model of random fields for other material parameters or structural parameters. The applicability and effectiveness of the proposed computational framework are evidently demonstrated through the numerical investigations on various practically motivated engineering structures.(4)Obviously, the simulation results are closer to reality when more parameters are considered with the random field. However, as the number of parameters considered increases, the computational effort increases exponentially. How to strike a valuable trade-off between them is an interesting area of future work.

## Figures and Tables

**Figure 1 materials-16-01188-f001:**
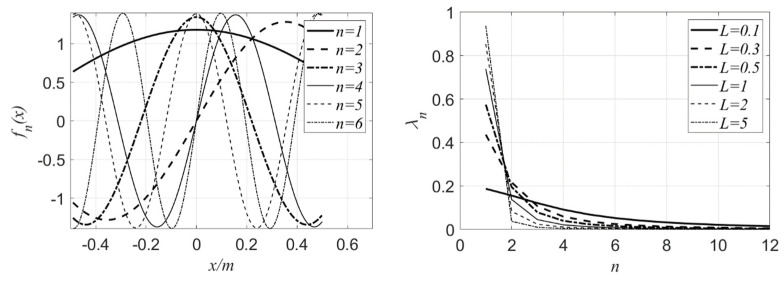
Eigenfunctions (**left**) and eigenvalues (**right**) of autocovariance matrix in the case of 1D.

**Figure 2 materials-16-01188-f002:**
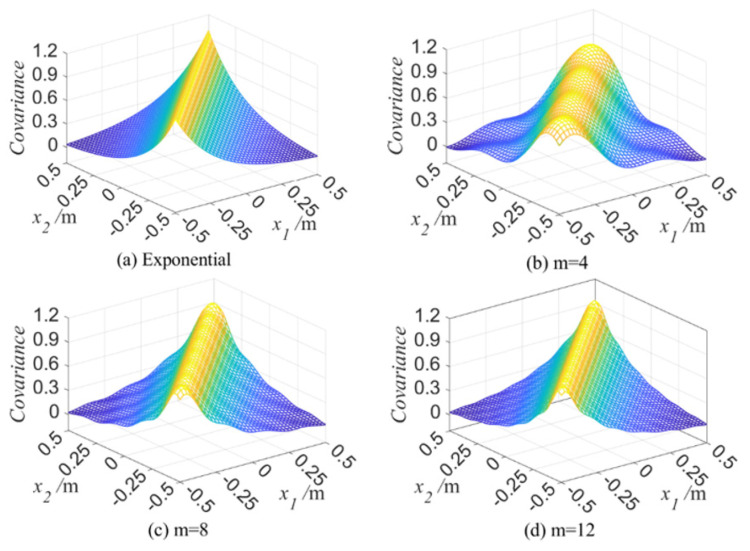
The autocorrelation function $C\left(x_{1},x_{2}\right)$ of 1D random variable when m=4,8,12, l=1m, and L=0.3m.

**Figure 3 materials-16-01188-f003:**
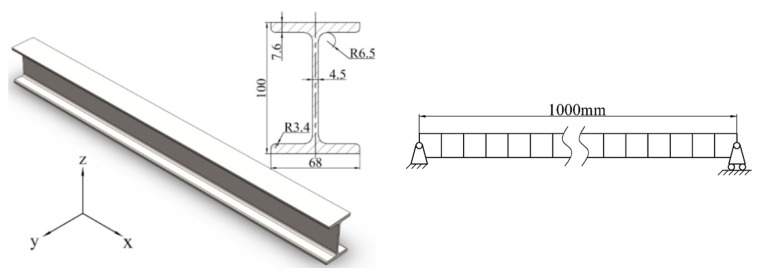
The geometric dimensions of No.10 I-beam (**left**) and its meshing elements (**right**) (unit: mm).

**Figure 4 materials-16-01188-f004:**
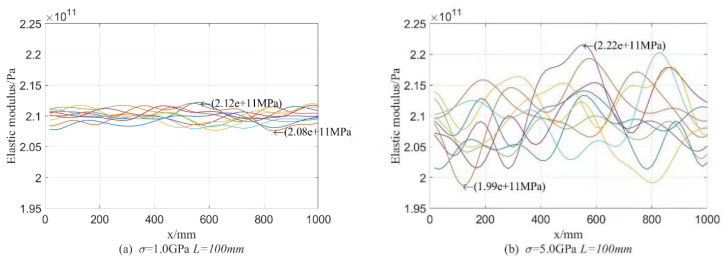
Random field of Young’s modulus E in the beam.

**Figure 5 materials-16-01188-f005:**
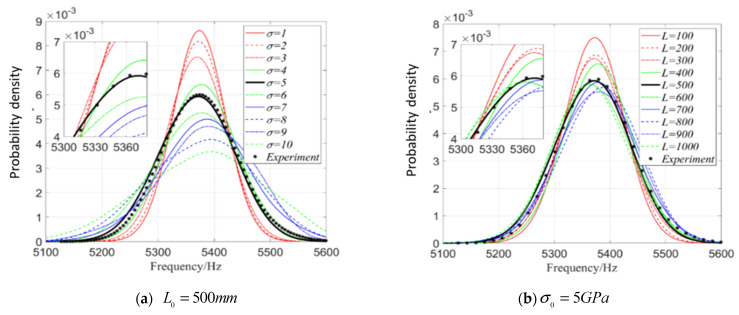
One-dimensional kernel density distribution function of $f_{2}$ when only considering random field of E.

**Figure 6 materials-16-01188-f006:**
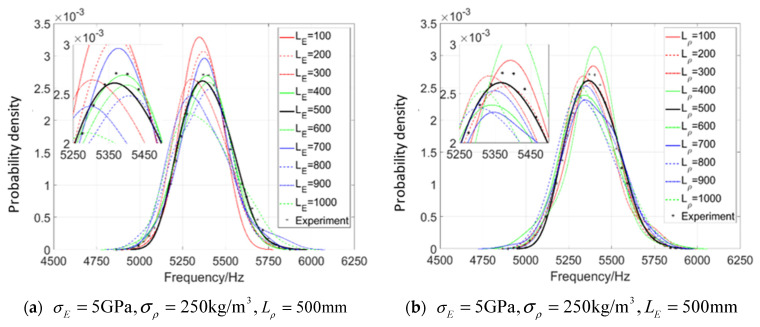
One-dimensional kernel density estimation of the second-order natural frequency when considering random fields E and ρ simultaneously.

**Figure 7 materials-16-01188-f007:**
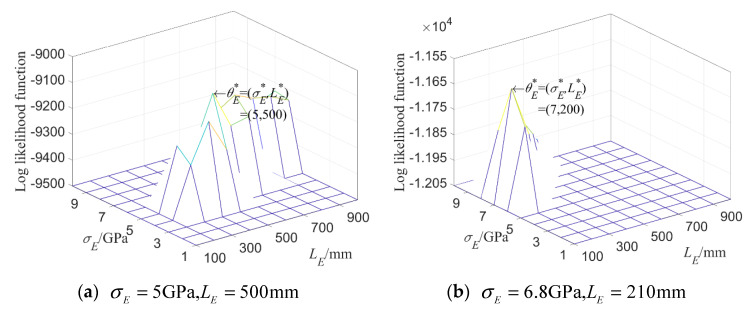
**Log likelihood function** of 8-dimensional kernel density distribution function of the first 8 orders’ natural frequency when only considering random field E.

**Figure 8 materials-16-01188-f008:**
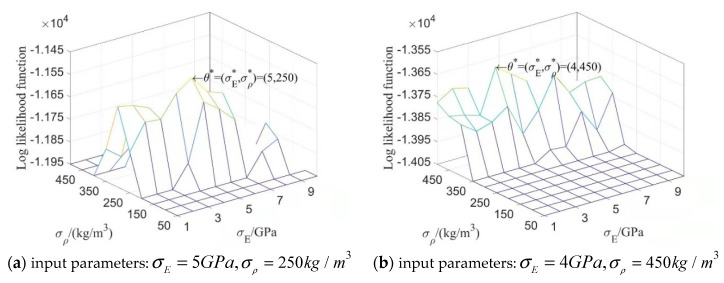
**Log likelihood function** of 8-dimensional kernel density distribution function of the first 8 orders’ natural frequencies when both E and ρ are random fields.

**Figure 9 materials-16-01188-f009:**
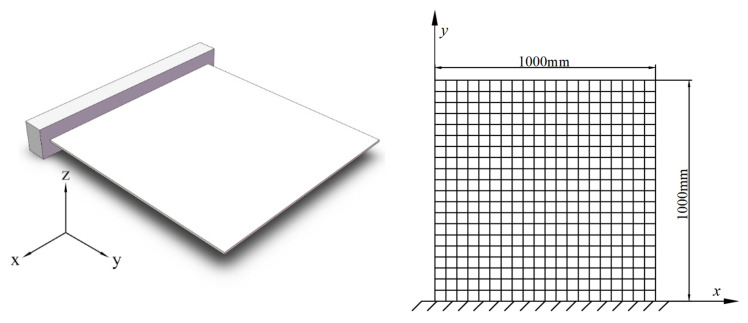
The plate fixed at one end (**left**) and its mesh elements (**right**).

**Figure 10 materials-16-01188-f010:**
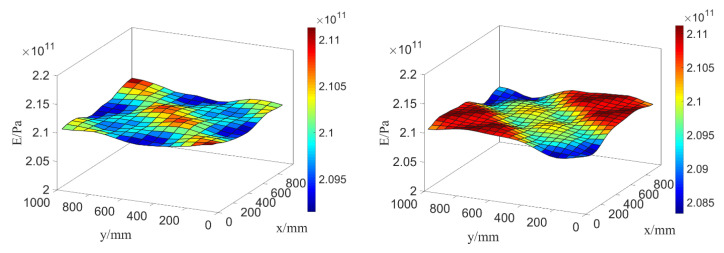
Random field distribution of E in the plate in the case of 15×15 elements (**left**) and 20×20 elements (**right**).

**Figure 11 materials-16-01188-f011:**
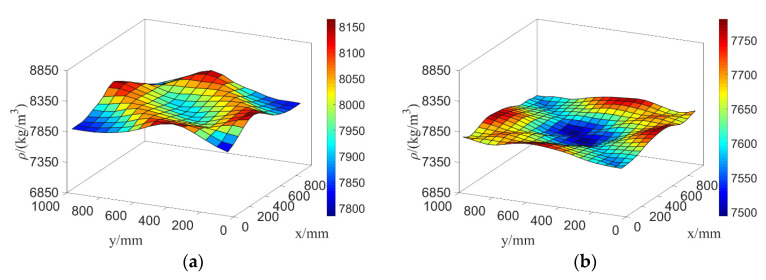
Random field distribution of ρ in the plate in the case of 15×15 elements (**a**) and 20×20 elements (**b**).

**Figure 12 materials-16-01188-f012:**
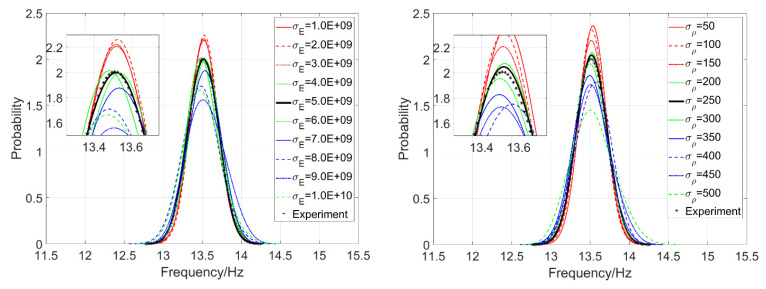
The two-dimensional kernel density estimation of $f_{2}$ when both E and ρ are random fields.

**Figure 13 materials-16-01188-f013:**
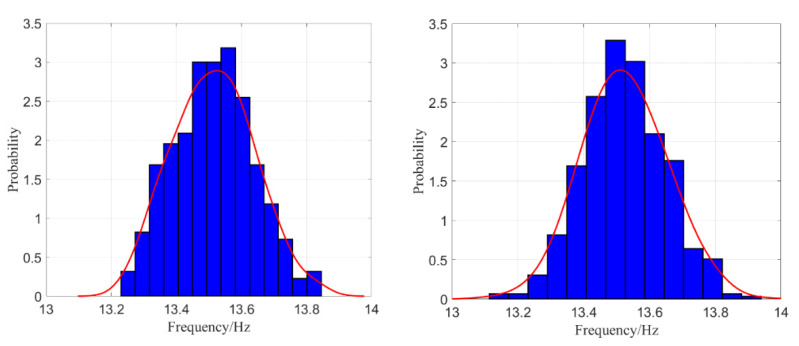
The histograms of $f_{2}$ when both E and ρ are random fields: 15×15 elements (**left**) and 20×20 elements (**right**).

**Figure 14 materials-16-01188-f014:**
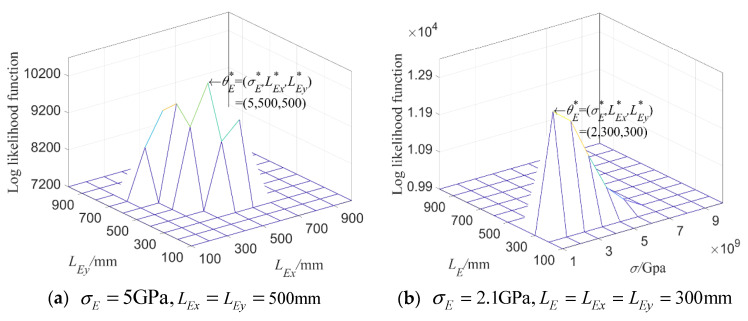
$\hat{L}\left(\theta \right)$ of the first 6 natural frequencies when inputting different parameter settings $\theta_{E}$.

**Figure 15 materials-16-01188-f015:**
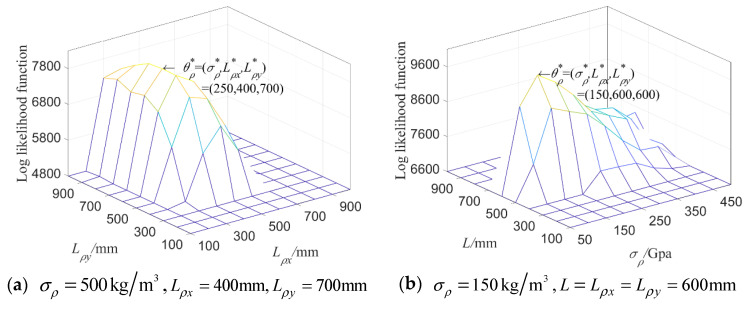
$\hat{L}\left(\theta \right)$ of the first 6 natural frequencies when inputting different parameter settings of random field ρ.

**Figure 16 materials-16-01188-f016:**
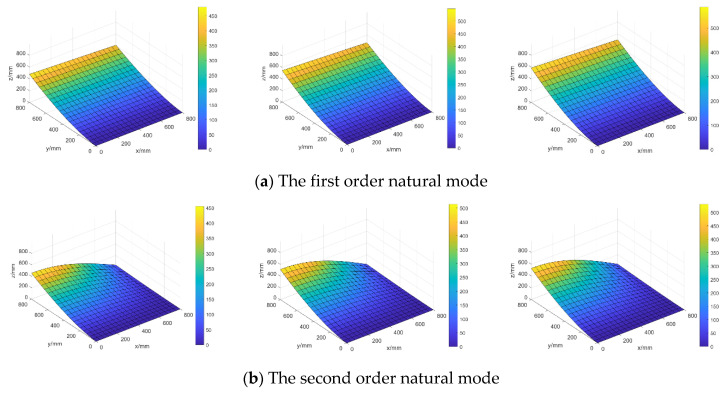
The distributions of lower bound (**left**), mean value (**middle**), and upper bound (**right**) of random natural modes when only considering random field E.

**Figure 17 materials-16-01188-f017:**
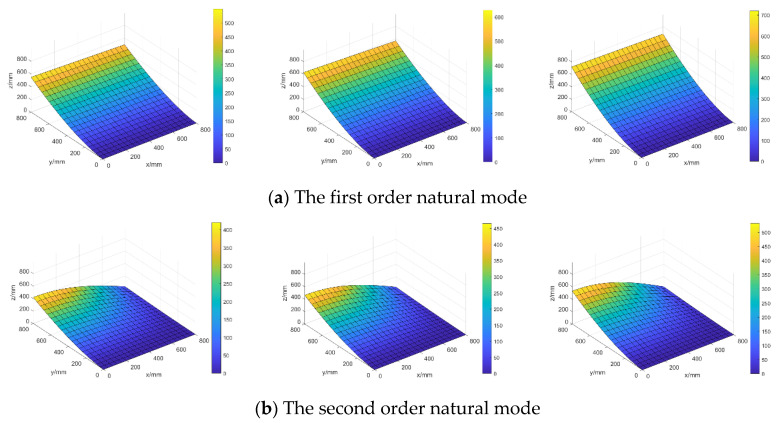
The distributions of lower bound (**left**), mean value (**middle**), and upper bound (**right**) of random natural modes when considering random fields E and ρ simultaneously.

**Figure 18 materials-16-01188-f018:**
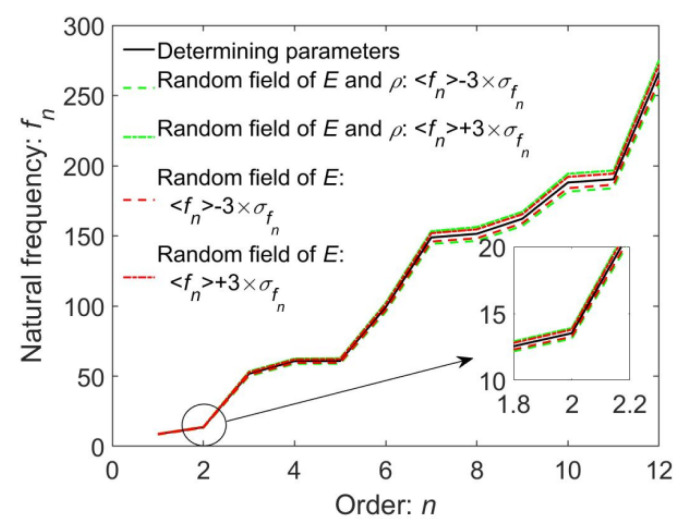
The value range of random natural frequencies for different random models computed by $\langle f_{n}\rangle \pm 3\times \sigma_{f_{n}}$.

**Table 1 materials-16-01188-t001:** The maximum likelihood estimation of test samples when considering random field E.

Input Parameter Setting of Test Samples of 500 Plates $\theta_{E}=\left(\sigma_{E},L_{Ex},L_{Ey}\right)$	The Maximal Value of Log Likelihood Function $\hat{L}\left(\theta_{E}\right)$	Parameter Setting Estimated from $\hat{L}\left(\theta_{E}\right)$			Mean Value and Mean Variance of the First Three Natural Frequencies Obtained from 500 Test Samples of 500 Plates					
		$\theta_{E}^{*}=\left(\sigma_{E}^{*},L_{Ex}^{*},L_{Ey}^{*}\right)$			$f_{1}$ (Hz)		$f_{2}$ (Hz)		$f_{3}$ (Hz)	
		$\sigma_{E}^{*}$	$L_{Ex}^{*}$	$L_{Ey}^{*}$	$\langle f_{1}\rangle$	$\sigma_{f_{1}}$	$\langle f_{2}\rangle$	$\sigma_{f_{2}}$	$\langle f_{3}\rangle$	$\sigma_{f_{3}}$
(5, 500, 500)	10,344.12	5	500	500	8.6185	0.0581	13.5240	0.0877	51.6952	0.3545
(6, 300, 700)	9190.03	6	300	700	8.6159	0.0659	13.5199	0.0969	51.6793	0.3957
(2, 100, 700)	13,712.95	2	100	700	8.6197	0.0131	13.5254	0.0183	51.7022	0.0776
(9, 430, 650)	8151.44	9	400	700	8.6204	0.1083	13.5271	0.1627	51.7069	0.6553

Herein, 〈·〉 and σ denote the mean value and mean variance of the random variables.

**Table 2 materials-16-01188-t002:** Comparison of the norm of natural modes for different random models.

	Computational Results	Norm of Random Natural Modes							
		$\Phi_{1}$		$\Phi_{2}$		$\Phi_{3}$		$\Phi_{4}$	
Random Models		$\langle \left\|\Phi_{1}\right\|\rangle$	$\sigma_{\left\|\Phi_{1}\right\|}$	$\left\|\Phi_{2}\right\|$	$\sigma_{\left\|\Phi_{2}\right\|}$	$\left\|\Phi_{3}\right\|$	$\sigma_{\left\|\Phi_{3}\right\|}$	$\left\|\Phi_{4}\right\|$	$\sigma_{\left\|\Phi_{4}\right\|}$
Deterministic model		6342.80	0	1635.00	0	399.12	0	296.67	0
Random field E		6346.67	94.82	1635.12	6.56	399.25	3.73	357.71	1010.50
Random fields E and ρ		6347.82	134.34	1635.43	7.71	399.21	3.98	477.17	1249.14

**Table 3 materials-16-01188-t003:** The mean values and mean variances of random natural frequencies for different random models.

	Computational Results	Mean Value and Mean Variance of Natural Frequencies							
		$f_{1}$		$f_{2}$		$f_{3}$		$f_{4}$	
Random Models		$\langle f_{1}\rangle$	$\sigma_{f_{1}}$	$\langle f_{2}\rangle$	$\sigma_{f_{2}}$	$\langle f_{3}\rangle$	$\sigma_{f_{3}}$	$\langle f_{4}\rangle$	$\sigma_{f_{4}}$
Deterministic model		8.62	0	13.52	0	51.67	0	60.68	0
Random field of E		8.62	0.0583	13.53	0.0885	51.69	0.3580	60.69	0.3914
Random fields of E and ρ		8.62	0.0906	13.52	0.1321	51.65	0.5651	60.62	0.6235

## Data Availability

Please contact the corresponding author for relevant data.

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
