# Peer review of "Uncertain Dynamic Characteristic Analysis for Structures with Spatially Dependent Random System Parameters"

_materials, 2023, doi:10.3390/ma16031188_

Round 1
Reviewer 1 Report
The authors presented a robust non-deterministic free vibration analysis for engineering 9 structures with random field parameters in the frame of stochastic finite element method. In general, the manuscript is very well written. However, there are some technical issues which must be addressed. I, therefore, recommended minor revision.
1. Plz present some quantitative information in the abstract.
2. I want to see some strong literature review presentation and problem statement discussion.
3. Plz shorten the section 3 heading.
4. Either use Tab for Table or use Table; similarly, use Fig, or Figure for figures throughout the manuscript.
5. Section 4.1.1.: What do you mean by four super parameters of Young’s modulus? It is not the correct way
6. Plz re-write the conclusion section as follows: Main objective, findings, advantages, limitations of the study and the future scope of the study.
7. Follow proper grammatical arrangement as stated below:
Check the following para how to apply grammatical correction: Use the past tense to report what happened in the past: what the authors did, what someone reported, what happened in an experiment, and so on. Use the present tense to express general truths, such as conclusions (drawn by the authors or by others) and facts not limited by time (including information about what the paper does or covers). Reserve the future tense for perspectives: what will be done in the coming months or years
Reviewer 2 Report
This study investigates the dynamic free response of structural components with randomness in their material properties. The effect of material uncertainty is studied for a simply supported beam as well as a clamped thin square steel plate. To improve this manuscript, the following comments shall be addressed:
General comments:
1- The methodology presented in this article is applied to address the randomness in the material properties, specifically the Youngs modulus and the density. Please elaborate a bit on whether or not this technique can be used to address other types of uncertainties observed in structural problems, such as the stiffness relative to the connections of structural members or support conditions at the boundaries.
2. How would an increase in the number of random parameters from one (E) to two (E and rho) affect the computation costs?
3. Please provide more explanation on what a certain correlation length in your models shows physically. Does it refer to the spatial distribution of that random variable in the model? Please elaborate a bit on the physical interpretation of the correlation length.
Minor comments:
4- Line 37, please check the sentence grammatically. A verb like “are” is missing.
5- In line 69, the expression “capable in” shall change to “capable of”.
6- Line 76, please use lowercase “a” after “Also”.
7- In the equation presented in line 202, please replace “m” with another symbol. The symbol “m” is used in the rest of the manuscript to refer to unit “meters”. Using the same symbol for two different parameters can make confusion.
8- Line 218, the correlation length is greater than the model length (L=2 or 5 m where l=1m). Please check. If needed add more discussion here.
9- Please cite the reference from which equations (29) and (30) are taken.
10- Regarding the presented example, please explain how the models are generated. Are all the analyses provided in the article based on numerical data? How are variables like E quantified by numerical tests? How are the numerical tests performed? Please clarify these topics and elaborate more on them.
11- In Figures 5, 6, and 12, what do the “Experimental” curves refer to? How are they obtained? (This comment may have some overlap with the previous one)
12- In Figures 7 and 8, the caption is missing the first word (they start with “ of”.). Please fix.
13- In Figures 10 and 11, the only difference between graphs (a) and (b) is the discretization scheme (15 elements vs 20). In this case, it is expected that both surfaces should be very similar to each other. However, the patterns are not identical. Please discuss the difference.
14- In table 1 the log-likelihood column contains large positive numbers. However, in figures 7 and 8, log-likelihood values are large negative values. Please check the data provided in this table.
15- In Figure 13, the y scale is the probability, which can not be larger than 1. However, the ordinate goes up to around 3.5. Please check your results.
16- Line 623, please consider using “Investigation of the random ..” instead of “Investigation to the random …”.
17- In Figures 16 and 17, the right edge of the plate is flat and undeformed. This could be due to the clamped and fixed support there. However, the z-values along this clamped edge are not zero in all graphs in both figures 16 and 17. Please explain the non-zero z values along the clamped edge or modify the figures.
18- In table 3, the upper left text is split into two lines (re- and sults). Please fix.
Reviewer 3 Report
The paper is well presented and contains original results. However, authors are encouraged to improve their work based on the following comments:
1) Authors may explain deficiencies or shortcomings of other studies to make a bridge to introducing the novelty of their work.
2) The novelty of this work must be more explained.
3) For general readers, authors are encouraged to discuss the possibility to use other kinds of methods (FEM or DQM) to solve this type of problems by considering the following recent works in the introduction: [(a) “Static bending and buckling analysis of bi-directional functionally graded porous plates using an improved first-order shear deformation theory and FEM”; (b) “On the vibrations of the Electrorheological sandwich disk with composite face sheets considering pre and post-yield regions”; (c) "Dynamic stability/instability simulation of the rotary size-dependent functionally graded microsystem”].
4) Some equations should be referenced.
5) In conclusion, give only main findings of your research with an appropriate value.
Round 2
Reviewer 2 Report
Thanks for revising the manuscript.
However, please make sure that:
1)- comments 10 and 11 are addressed. I did not see much elaboration in paragraphs 1 and 2 in section 4.1.2.
2)- comment 16 is also addressed (“Investigation of …”).
Besides, the conclusion seems to be in section 5, not 4.